# Lipopolysaccharide Tolerance Enhances Murine Norovirus Reactivation: An Impact of Macrophages Mainly Evaluated by Proteomic Analysis

**DOI:** 10.3390/ijms24031829

**Published:** 2023-01-17

**Authors:** Jiradej Makjaroen, Pornpimol Phuengmaung, Wilasinee Saisorn, Suwasin Udomkarnjananun, Trairak Pisitkun, Asada Leelahavanichkul

**Affiliations:** 1Center of Excellence in Systems Biology, Faculty of Medicine, Chulalongkorn University, Bangkok 10330, Thailand; 2Center of Excellence in Translational Research in Inflammation and Immunology (CETRII), Faculty of Medicine, Chulalongkorn University, Bangkok 10330, Thailand; 3Department of Microbiology, Faculty of Medicine, Chulalongkorn University, Bangkok 10330, Thailand; 4Division of Nephrology, Department of Medicine, Faculty of Medicine, Chulalongkorn University, Bangkok 10330, Thailand

**Keywords:** sepsis immune exhaustion, lipopolysaccharide tolerance, macrophages, murine norovirus, viral reactivation

## Abstract

Because of endotoxemia during sepsis (a severe life-threatening infection), lipopolysaccharide (LPS) tolerance (the reduced responses to the repeated LPS stimulation) might be one of the causes of sepsis-induced immune exhaustion (the increased susceptibility to secondary infection and/or viral reactivation). In LPS tolerance macrophage (twice-stimulated LPS, LPS/LPS) compared with a single LPS stimulation (N/LPS), there was (i) reduced energy of the cell in both glycolysis and mitochondrial activities (extracellular flux analysis), (ii) decreased abundance of the following proteins (proteomic analysis): (a) complex I and II of the mitochondrial electron transport chain, (b) most of the glycolysis enzymes, (c) anti-viral responses with Myxovirus resistance protein 1 (Mx1) and Ubiquitin-like protein ISG15 (Isg15), (d) antigen presentation pathways, and (iii) the down-regulated anti-viral genes, such as *Mx1* and *Isg15* (polymerase chain reaction). To test the correlation between LPS tolerance and viral reactivation, asymptomatic mice with and without murine norovirus (MNV) infection as determined in feces were tested. In MNV-positive mice, MNV abundance in the cecum, but not in feces, of LPS/LPS mice was higher than that in N/LPS and control groups, while MNV abundance of N/LPS and control were similar. Additionally, the down-regulated *Mx1* and *Isg15* were also demonstrated in the cecum, liver, and spleen in LPS/LPS-activated mice, regardless of MNV infection, while N/LPS more prominently upregulated these genes in the cecum of MNV-positive mice compared with the MNV-negative group. In conclusion, defects in anti-viral responses after LPS tolerance, perhaps through the reduced energy status of macrophages, might partly be responsible for the viral reactivation. More studies on patients are of interest.

## 1. Introduction

Sepsis is a potentially life-threatening condition in response to severe infection regardless of the organismal causes (bacteria, viruses, fungi, and parasites) [1]. While there is a surprisingly similar syndrome in sepsis from different causes, the immune responses in sepsis are possibly different, which are crudely categorized into hyperinflammation and immune exhaustion (immune paralysis) [2]. 

Hence, the balance in sepsis immune responses is complex, and a proper immune modulation, such as the anti-inflammation during sepsis-hyperinflammation and the boost-up of immune responses in immune exhaustion status, might be beneficial [3,4,5,6,7,8]. Indeed, enhanced susceptibility to secondary infection after sepsis (mostly by bacteria or fungi) is well-known as obvious supportive evidence for sepsis immune exhaustion [9]. Likewise, reactivation of the dormant virus, including cytomegalovirus (CMV), Epstein–Barr (EBV), herpes simplex (HSV), and human herpes virus-6 (HHV-6), in patients with sepsis is also mentioned [10,11]. Interestingly, CMV reactivation is possibly an interesting biomarker of sepsis immune exhaustion [12]. However, the hyperimmune responses and anti-inflammatory immune exhaustion in sepsis seem to occur in the same patients simultaneously, and tips the balance of the responses might determine the direction of clinical sepsis manifestation [13]. As such, a surgical sepsis induction in mice causes the emergence of the dormant *Candida* in the reticuloendothelial system (lungs, livers, and spleens) and into the blood (Candidemia) as early as 6 h post-surgery, indicating a defect of microbial control at an early phase of sepsis [14], which is highlighting a possibly important sepsis-induced immune exhaustion. 

Due to the improved supportive care during hyper-inflammatory sepsis [15], secondary infection after sepsis might be an emerging problem in sepsis treatment. Although defects of microbial control during immune exhaustion are caused by several factors, including the death of several immune cells (especially apoptosis), myeloid-derived suppressor cells, enhanced regulatory T cells, and lipopolysaccharide (LPS) tolerance (the decreased immune responses following the second dose or prolonged LPS stimulation) [16,17], data on impacts of LPS tolerance on infection is still scarce. Because (i) LPS is a major molecule of the cell wall of Gram-negative bacteria (the most abundant organism in the gut), (ii) the translocation of LPS from the gut into the blood circulation (gut leakage or leaky gut) is common in sepsis, partly due to the relatively low oxygen perfusion in the gut (distributive shock), resulting in sepsis-induced endotoxemia [18,19,20], and (iii) a natural adaptation to the repeated LPS stimulations by reducing the cytokine responses (LPS tolerance), which might be inadequate for microbial control [21,22]; thus, LPS tolerance might be, at least in part, important in sepsis-induced secondary infection and viral reactivation. However, data on the correlation between LPS tolerance and viral reactivation are still very sparse. Additionally, the underlying mechanisms of the reduced responses in LPS tolerance, especially in monocytes or macrophages, are still unclear, which might consist of epigenetic alterations, chromatin remodeling, and interferences on cell energy status [23]. Then, the proteomic analysis focusing on cell energy status, immune responses to viruses, and microbial control processes might be helpful for the understanding of LPS tolerance and viral reactivation. 

In parallel, murine norovirus (MNV) is a single-stranded RNA virus that causes acute watery diarrhea with a self-limiting natural course within 48 h to 72 h after infection, but the mice can continuously shed these viruses, and the symptoms can re-emerge in immunocompromised hosts [24]. Hence, MNV is an interesting virus to test the viral reactivation after LPS tolerance in an animal model that will directly support the correlation between viral reactivation and LPS tolerance. Here, proteomic analysis in LPS tolerance macrophages (twice-stimulated LPSs) compared with single LPS-stimulated cells using RAW264.7 (murine macrophage cell line) with the investigation of the LPS tolerance effect in MNV-infected versus control mice was performed. 

## 2. Results

### 2.1. The Reduction of Cell Energy Status and Inflammatory Responses in Macrophages with LPS Tolerance

Due to the correlation between several macrophage activities (cell energy status, inflammatory signaling, viral replication, and antigen presentation) in hyperinflammation and immune exhaustion during sepsis [25,26], proteomic liquid chromatography–tandem mass spectrometry (LC-MS/MS) analysis in LPS-activated RAW264.7 cells with a single stimulation (N/LPS) and LPS tolerance (LPS/LPS) was performed in 3 isolated replications as the representatives or hyper- versus anti-inflammatory phase of sepsis, respectively (Figure 1A). The number of identified proteins in LPS/LPS macrophages relative to N/LPS macrophages was 2444 peptides, as illustrated in the volcano plot (Figure 1B) that consisted of 565 significantly upregulated proteins and eight-hundred down-regulated groups (some peptides were in the same proteins). With DAVID Bioinformatics tools, the classification of these proteins with the top 10 relevant pathways indicated that most proteins were members of metabolic processes, followed by the immune response pathways and the signal transduction (Figure 1C). As such, most of the proteins in complex I (CI) and complex II (CII) of the mitochondrial electron transport chain was lower in LPS tolerance (LPS/LPS) macrophages compared with N/LPS macrophages, despite an increase in most of the protein in complex V and the tricarboxylic acid (TCA) cycle of mitochondrial oxidative phosphorylation (OXPHOS), perhaps as a compensation for a defect in CI and CII (Figure 2A, upper part on the left side). In parallel, there was a reduction in mitochondrial function and glycolysis activity of LPS/LPS macrophages compared with N/LPS cells, as indicated by the oxygen consumption rate (OCR) from the extracellular flux analysis (Figure 2B) and a decrease in the important rate-limiting glycolytic enzymes, including hexokinase 2 (Hk2) and phosphofructokinase (liver- and platelet-type, Pfkl and Pfkp) (Figure 2A, right side). Additionally, there was higher expression of several enzymes in glutamine metabolism (Got1, Got2, Gls, and Glud1) in LPS/LPS macrophages over the N/LPS cells (Figure 2A, left lower part). 

Because of a possible correlation between a cell’s energy status and macrophage responses [26,27,28], proteins of inflammatory responses, viral cycle, and antigen presentation were also explored from the proteomic analysis (Figure 3). Although there was an increase in several cell surface receptor-associated proteins, LPS/LPS macrophages demonstrated a reduction in most of the downstream signals, including (i) nuclear factor kappa B (NFκB, an important transcriptional factor used for the production of several pro-inflammatory molecules), including NFκB1 (p50), NFκB2 (p52), Rela (p65), and Rel, and (ii) mitogen-activated protein kinase (MAPK, an important downstream protein of the MyD88-dependent pathways), including Map2k4, Mapk1, and Mapk3, leading to a reduction in other transcriptional factors, such as activator protein-1 (AP-1), (iii) CCAAT/enhancer-binding protein β (Cebpb), the transcriptional factor-targeting pro-inflammatory genes, and (iv) CARD9-BCL10-MALT1 (CBM) signalosome, a complex mediator of NF𝜅B signaling from Caspase recruitment domain family member 9 (Card9), B-cell lymphoma/leukemia 10 (Bcl10), and mucosa-associated lymphoid tissue lymphoma-translocation gene 1 (Malt1) (Figure 3 upper). In contrast, the negative regulators of the TLR4 signaling pathway, including Tollip and Tnip1, were significantly upregulated in LPS tolerance macrophages (Figure 3 upper), perhaps for compensation of reduced pro-inflammatory downstream signals. Meanwhile, the cytosol receptor of RNA viruses, including (i) melanoma differentiation-associated 5 (MDA5), encoded from interferon induced with helicase C domain 1 (Ifih1) gene, and (ii) Laboratory of genetics and physiology 2 (LGP2), encoded by Dhx58 (ATP-dependent RNA helicase DExH-box helicase 58), were lower in LPS/LPS macrophages with higher TNIP1 (TNFAIP3 Interacting Protein 1, a negative NFκB regulator) (Figure 3 lower at the left side), which possibly resulted in the lack of type I and III interferons (IFNs) for the viral control [29,30]. For the antigen presentation, there were both up- and down-regulated proteins of this process in LPS/LPS macrophages (Figure 3, lower right side); however, the requirement of ATP for proteasome to generate antigenic peptides might interfere with the antigen presentation process of macrophages, leading to limited functions of cytotoxic T cells and viricidal activity [31]. Hence, the data on proteomic analysis demonstrated a possible defect of LPS tolerance macrophages on the responses against viral infection through a reduction in cellular energy status, inflammatory responses, cytosol viral recognition, and antigen presentation processes. Although there were up- and down-expressions of interferon-inducible proteins in LPS tolerance macrophages compared with N/LPS macrophages (Figure 3 upper, lower left), some interferon signatures acting as an intracellular sensor of viral RNA, including Ifih1 (MDA5) and Dhx58 (LGP2), were down-regulated in endotoxin tolerance (Figure 3 lower, left side), which possibly facilitates viral production during LPS tolerance. Subsequently, the gene expression of pro-inflammatory cytokines, NFκB synthesis (*Rela*), and anti-viral processes was explored in LPS-stimulated macrophages. In comparison with N/LPS macrophages, LPS/LPS cells demonstrated down-regulated genes of pro-inflammation (*Tnfa* and *Il1b*), NFκB synthesis (*Rela*), viral control (*Mx1*, myxovirus resistance protein 1), and *Isg15* (interferon-stimulated gene 15) (Figure 4A–E). Interestingly, there was an upregulation of all of these genes in N/LPS macrophages, possibly due to the activation by inflammatory cytokines (upregulated *Tnfa* and *Il1b*) (Figure 4A,B) that activate downstream NF𝜅B (upregulated *Rela*) (Figure 4C), resulting in enhanced anti-viral molecules (upregulated *Mx1* and *Isg15*) (Figure 4D,E). 

### 2.2. LPS-Tolerant Mice Demonstrated More Prominent Diarrhea with More Profound Expression of Murine Norovirus Than Other Groups

The MNV-positive or MNV-negative mice were divided into three groups to investigate the impact of LPS tolerance on viral replication in vivo, including the control (N/N), LPS response (N/LPS), and LPS tolerance (LPS/LPS) before sacrificing and collecting samples (feces, blood, and ceca) after 1 h of the protocol (Figure 5A). In MNV-negative mice, both N/LPS and LPS/LPS induced normal fecal consistency (Figure 5B,C). Despite LPS-enhanced intestinal motility [32], the selected doses of LPS in our experiments were not high enough to induce diarrhea in healthy mice [33]. In contrast, the feces of MNV-positive mice had a soft consistency after a single LPS injection and then became diarrhea after LPS was administered twice (LPS tolerance) (Figure 5B,C). Because the mucosal injury during diarrhea might damage the intestinal tight junction enough to cause a leaky gut [34,35,36], FITC-dextran assay for gut barrier evaluation was performed. Indeed, the severity of leaky gut was most severe in MNV-positive mice with LPS tolerance followed by a single LPS stimulation, while the MNV-positive control and LPS stimulation (N/LPS and LPS/LPS) in MNV-negative mice were non-different than the normal (N/N) MNV-negative control mice (Figure 5D). Despite the most severe leaky gut in LPS/LPS of MNV-positive mice, serum cytokines (TNF-α, IL-1β, and IL-6, but not IL-10) of these mice were lower than N/LPS in the MNV-positive group (Figure 5E–H), supporting the characteristics of LPS tolerance in these mice (i.e., lower inflammatory responses due to the repeated doses of LPS) [22,37]. Notably, the levels of serum cytokines of MNV-positive and MNV-negative mice with the same stimulation by N/LPS or LPS/LPS were similar (Figure 5E–H), despite the difference in leaky gut severity (Figure 5D), supporting a minimal impact of MNV in these mouse models and a possible leaky gut with a non-significant pro-inflammatory effect. 

Because diarrhea in MNV-positive mice with either N/LPS or LPS/LPS, at least in part, might be due to the viral reactivation, the burdens of MNV were evaluated. In fecal contents of MNV-positive mice, MNV abundance was not different among control (N/N), N/LPS, and LPS/LPS, when compared with the MNV-negative mice (Figure 6A). However, MNV abundance was highest in the cecum of MNV-positive mice than in other groups with a non-different cecal MNV between control and N/LPS MNV-positive mice (Figure 6B). In parallel, the expression of inflammation-associated genes (*Ifih1*), used for the production NFκB (a pro-inflammatory transcriptional factor), and anti-viral genes (*Mx1* and *Isg15*) in the cecum of LPS/LPS MNV-positive mice was lower than MNV-positive N/LPS (Figure 6C–E). Interestingly, in the cecum of MNV-negative mice, N/LPS and LPS/LPS demonstrated up- and down-regulated *Mx1*, respectively, and the *Mx1* expression of LPS/LPS MNV-negative mice was also lower than N/LPS (Figure 6D), which implied a correlation between LPS tolerance and the reduced anti-viral activities. Additionally, the N/LPS stimulation might be able to enhance the systemic anti-viral activities since *Mx1* and *Isg15* expressions were greatly elevated in the livers and spleens of N/LPS MNV-negative mice (Figure 6F–I). Notably, the liver *Mx1*, but not *Isg15* nor these genes in the spleen of LPS/LPS MNV-negative mice, was downregulated when compared with MNV-negative controls (Figure 6F–I). The upregulation of *Mx1* and *Isg15* in the liver of N/LPS MNV-positive mice was lower than in N/LPS MNV-negative mice (Figure 6F,G), while *Mx1* and *Isg15* in the spleen of N/LPS-activated mice with MNV-negative versus MNV-positive were similar (Figure 6H,I), implying a difference between the anti-viral activities in different organs. 

## 3. Discussion

### 3.1. LPS Tolerance Interfered with Several Macrophage Activities Partly through a Reduction in Cell Energy

The analyses of a single versus a twice-stimulated LPS in macrophages were performed to demonstrate the difference between the regular versus repeated responses against LPS, a microbial molecule from Gram-negative bacteria, which was clinically demonstrated as endotoxemia in several conditions [38,39,40]. The most abundant proteins in LPS tolerance (LPS/LPS) macrophages compared with the single LPS-activated macrophages (N/LPS) from the DAVID Bioinformatics tool were the members of metabolic processes, indicating a potential metabolic reprogramming during macrophage responses against infection, as supported by previous publications [28,41,42]. The less abundance of proteins in complexes I (CI) and II of the mitochondrial electron transport chain in LPS/LPS macrophages than N/LPS cells might be compensated by an increase in most of the protein in complexes III – V and in the TCA cycle of mitochondrial OXPHOS. The defect in CI of the electron transport chain might also be responsible for the reduced mitochondrial function in LPS/LPS macrophages as evaluated by the OCR of extracellular flux analysis. Despite the up- and down-regulated proteins (mostly down-regulation) in the glycolytic pathway of LPS/LPS cells compared with N/LPS macrophages, the important rate-limiting enzymes, including hexokinase 2 (Hk2) and phosphofructokinase (liver- and platelet-type, Pfkl and Pfkp) in LPS/LPS macrophages were lower than N/LPS groups, which might be responsible for the reduced glycolysis (the ECAR of extracellular flux analysis) and a compensatory increase in nearly all proteins in the TCA cycle of the mitochondrial matrix (Figure 2). These overall changes might be responsible for the reduced cell energy status of LPS tolerance macrophages, which is supportive of reduced cellular ATP mentioned in a previous publication [43]. Interestingly, the higher expressions of several enzymes in glutamine metabolism (Got1, Got2, Gls, and Glud1) in LPS/LPS macrophages over the N/LPS cells might be another compensation for the lower cell energy status by promoting the conversion of glutamine into α-ketoglutarate (a key metabolite in the TCA cycle). As such, the enhanced glutamine metabolism might be an attempt to enhance the production of nicotinamide adenine dinucleotide with hydrogen (NADH) and flavin adenine dinucleotide with hydrogen (FADH2) that are further used for the synthesis of adenosine triphosphate (ATP, a molecule providing energy for many processes of living cells) in the TCA cycle [26,27].

Notably, cell exhaustion after a potent response against the previous LPS stimulation before a repeated dose of LPS was possible as there was an increase in glycolysis activity after a single LPS stimulation (extracellular flux analysis) that might need several substrates and ATP for the production of profound inflammatory responses (cytokines and other signals) [44]. In short, the extracellular flux analysis demonstrated a reduction in both glycolysis and mitochondrial activities in LPS tolerance macrophages, while there was an increase in glycolysis activity without an alteration in mitochondrial function in the single LPS-stimulated cells, supporting previous publications [5,45,46,47]. Subsequently, the reduced macrophage cell energy in LPS tolerance might be correlated with the reduction of several signals, including NFκB, CBM signalosome, MAPK, inflammasome, and several genes of viricidal activities (interferon production and antigen presentation), which might be correlated with viral reactivation after sepsis. Among all responses, Mx1 and Isg15 were important for the interferon-stimulated genes (Isg) for the viricidal activities (a central gene in the host’s anti-viral response) that significantly downregulated after LPS tolerance macrophages. Moreover, reduced cytokine production after LPS tolerance has been shown to be a factor that facilitates secondary infections [21,22]. From another point of view, LPS tolerance allows the host to survive repeated Gram-negative bacterial infections by reducing the lethal hyper-inflammatory responses [48]; however, the reduced responses might lead to increased proliferation of the organisms because of the well-known importance of the inflammatory process for the microbial control [49]. The possible viricidal activity after a profound acute inflammation from a single LPS stimulation is also reported [50]. Indeed, the upregulated anti-viral proteins during a non-specific hyper-inflammation in the single LPS-activated macrophages might be responsible for (i) a less frequent mixed bacteria–virus infection in an early phase of bacterial sepsis and (ii) a higher prevalence of post-sepsis viral reactivation in the host with immune exhaustion than during hyper-inflammatory sepsis [12,51]. Hence, the use of LPS tolerance for an anti-inflammatory strategy possibly needs good microbial control in the host [52]. Although the beneficial effects of LPS preconditioning before cecal ligation and puncture sepsis induction on reduced inflammation with less blood bacterial burdens are mentioned [53], several publications indicate worsening mortality of LPS preconditioning before sepsis induction [22,54]. Perhaps, these inconsistencies are partly due to differences in the use of antibiotics, protocols of LPS administration, host conditions, and timing between the sepsis induction after LPS administration. More studies on the proper use of LPS tolerance-induced immune modulation for sepsis attenuation are of interest. 

### 3.2. LPS Tolerance Enhanced the Reactivation of MNV Infection

Infection by MNV (the RNA virus) is a well-known virus causing enteritis in rodents with some different strain-dependent characteristics, as the CR6 strain is restricted to the intestines and the CW3 strain systemically replicates; however, MNV is usually cleared in the immune-competent mice, causing an asymptomatic carrier [55]. Here, MNV-S99 was used due to the diarrhea-inducible property (local intestinal infection) with a natural asymptomatic dormant history in the immunocompetent mice [24], and the asymptomatic-infected mice were tested for viral reactivation through the severity of diarrhea and viral abundance (in feces and cecum). Here, N/LPS in MNV-negative mice did not induce diarrhea that was different from previous reports of increased gut motility from LPS-induced high pro-inflammatory cytokines [56], perhaps due to the differences in doses and types of the LPS. On the other hand, there was no report of diarrhea in LPS tolerance [57], which is similar to our LPS/LPS-administered MNV-negative mice, partly due to the lesser amount of inflammatory responses and lower cytokine production in LPS tolerance when compared with a single LPS stimulation. Interestingly, N/LPS and LPS/LPS activation in MNV-positive mice caused soft stool and diarrhea, respectively, with an increase in MNV abundance in cecal tissue (but not in feces). As such, MNV was detectable after the LPS stimulation protocols, but only in the feces of MNV-positive mice and not in MNV-negative hosts; however, the fecal viral abundance after N/LPS and LPS/LPS stimulations did not differ from the MNV-positive control (N/N) group. Because the MNV virus is an intracellular organism that might be presented in feces through the presence of several cells in fecal contents, especially the immune cells and enterocytes, the non-detectable MNV in diarrheal contents might be due to too few immune cells in the diarrheal contents (non-inflammatory diarrhea). Nevertheless, MNV in the cecum of LPS/LPS MNV-positive mice was higher than N/LPS injection, implying a possible defect in anti-viral activities from LPS tolerance, while a single LPS stimulation did not change cecal MNV burdens. Indeed, LPS/LPS downregulated expression of some important anti-viral genes (*Mx1* and *Isg15*) in all organs (cecum, liver, and spleen), similar to the decrease in these genes on macrophages (proteomic analysis). Notably, the liver and spleen are the organs with a high abundance of macrophages. The decreased expression of *Mx1* and *Isg15* in the spleen and liver of LPS/LPS-administered-MNV-positive mice suggests a possible reduction in systemic anti-viral activities. Further tests on the impact of LPS tolerance using other viruses with systemic infection are of interest. However, MNV infection did not alter the characteristics of N/LPS and LPS/LPS models in terms of the level of serum cytokines, implying a lesser impact of MNV on these models of systemic inflammation. For N/LPS MNV-positive mice, mouse feces were softened without overt diarrhea, possibly due to the LPS-enhanced intestinal motility, which was not detected in N/LPS MNV-negative mice. Despite the non-difference in systemic cytokine responses after N/LPS in MNV-negative versus MNV-positive mice, MNV-positive intestines seem to have a higher susceptibility to LPS-induced diarrhea than the MNV-negative gut supported an effect of MNV-positive in the models with intestinal inflammation. Due to the latent MNV infection in some mouse facilities, the use of MNV-positive mice for the models with intestinal inflammation might not appropriately mimic the human conditions, while the use of MNV-positive mice for the systemic inflammatory models might be acceptable.

Nevertheless, the latent MNV infection is an interesting virus to demonstrate viral reactivation in mice because most of the viruses with latent infection in humans, especially the viruses in the herpes family, do not infect the immunocompetent rodent. Here, our in vivo data supported a possible viral reactivation of cecal MNV with LPS tolerance, which might partly be responsible for the more severe diarrhea in LPS/LPS-activated MNV-positive mice over the MNV-negative mice. We hypothesized that LPS tolerance might partly be responsible for the viral reactivation during sepsis together with other factors of sepsis-induced immune exhaustion. The use of MNV for the viral reactivation topic is uncomplicated, and further experiments focusing on the impacts of MNV in several mouse models are of interest. 

### 3.3. Clinical Aspect and Future Experiments

Our data indicated possible defects in anti-viral activity in LPS tolerance macrophages using proteomic analysis through the reduction in proteins of interferon synthesis and antigen presentation processes. The defect in microbial control is partly due to the cell’s reduced energy status, which was supported by MNV reactivation and diarrheal manifestation in mice with twice-stimulated LPS, as concluded in Figure 7. Because of the possible chronic endotoxemia in several conditions (such as uremia, obesity, and sepsis), defects in microbial control and viral reactivation in patients with these conditions are possible. Moreover, LPS tolerance might be an additional mechanism responsible for secondary infection in patients with sepsis-induced immune exhaustion. Hence, the measurement of serum LPS and/or the ex vivo test on isolated macrophages to clinically identify LPS tolerance might be of interest. Hence, more clinical studies are warranted. 

## 4. Materials and Methods

### 4.1. Cell Culture and Stimulations

Murine macrophage-like cells (RAW264.7 and TIB-71), purchased from the American Type Culture Collection (ATCC, Manassas, VA, USA), were maintained in Dulbecco’s Modified Eagle’s Medium (DMEM, Cytiva HyClone) supplemented with 10% Fetal Bovine Serum (FBS) in a humidified incubator at 37 °C with 5% CO_2_. To generate LPS tolerance (LPS/LPS), the cells at 1 × 10^6^ cells/well were firstly stimulated with 100 ng/mL LPS (*Escherichia coli* 026:B6) (Sigma-Aldrich^®^, St. Louis, MO, USA) for 24 h, washed with PBS, and then re-stimulated with the same dose of LPS for another 24 h. For LPS response (N/LPS), RAW264.7 were grown in culture media for 24 h, then washed with PBS, and incubated with LPS 100 ng/mL for 24 h. The control group (N/N) was performed by growing RAW264.7 in media for 24 h, washing with PBS, and incubating in the media again for 24 h. Subsequently, cell pellets and supernatants were collected for further analyses, including proteomic analysis, cytokine measurement, and polymerase chain reaction (PCR). 

### 4.2. Mass Spectrometry Proteomic Analysis

Both N/LPS and LPS/LPS cells were processed for in-solution digestion, as described in the previous study [21]. Then, peptides from LPS response and LPS tolerance macrophages were labeled with 130 and 131 Tandem Mass Tag (TMT) reagents (Thermo Fisher Scientific, San Jose, CA, USA) according to the manufacturer’s instructions. The pooled peptides were fractionated using a high pH reversed-phase peptide fractionation kit (Thermo Fisher Scientific, San Jose, CA, United States). Liquid chromatography–tandem mass spectrometry (LC-MS/MS) analysis of samples was performed on an EASY-nLC1000 system coupled to a Q-Exactive Orbitrap Plus mass spectrometer equipped with a nano-electrospray ion source (Thermo Fisher Scientific, San Jose, CA, United States). The mass-spectrometry (MS) raw files were searched against the Mouse Swiss-Prot Database with a list of common protein contaminants. The search parameters were set for the following fixed modifications: carbamidomethylation of cysteine (+57.02146 Da), as well as TMT 6 plex of N-termini and lysine residue (+229.162932 Da) and variable modification: oxidation of methionine (15.99491 Da). The reporter ion intensity ratios of LPS/LPS vs. N/LPS were transformed to log2 (LPS/LPS vs. N/LPS). The *p*-values were calculated with Student’s t-test based on the triplicate log2 (LPS/LPS vs. N/LPS) against 0. The proteins with a *p*-value < 0.05 were considered significant proteins, and these proteins were subjected to the online DAVID Bioinformatics Resources 6.8 to investigate the enriched biological processes. The mass spectrometry proteomics data have been deposited to the ProteomeXchange Consortium (http://proteomecentral.proteomexchange.org) via the PRIDE partner repository with the dataset identifier PXD027471 and accessed on 31 January 2022.

### 4.3. Extracellular Flux Analysis

Seahorse XFp Analyzers (Agilent, Santa Clara, CA, USA) were used to determine the energy status of the cells (extracellular flux analysis), with oxygen consumption rate (OCR) and extracellular acidification rate (ECAR) representing mitochondrial function (respiration) and glycolysis activity, respectively, to follow previous publications [20,58,59,60,61]. Briefly, the macrophages (1 × 10^5^ cells/well) 1 h after the stimulations (N/N, N/LPS, and LPS/LPS) were incubated in Seahorse media (DMEM complemented with glucose, pyruvate, and L-glutamine, Agilent, 103575–100) before activation by different metabolic interference compounds, such as oligomycin, carbonyl cyanide-4-(trifluoromethoxy)-phenylhydrazone (FCCP), and rotenone/antimycin A for OCR evaluation or glucose, oligomycin, and 2-Deoxy-d-glucose (2-DG) for ECAR evaluation. The graphs of OCR and ECAR were demonstrated. 

### 4.4. Animals and Animal Model

The protocol was approved by the Institutional Animal Care and Use Committee of the Faculty of Medicine, Chulalongkorn University, Bangkok, Thailand (CU-ACUP No. 021/2562) according to the National Institutes of Health’s (NIH) criteria. Eight-week-old male C57BL/6 mice were used in this study (Nomura Siam, Pathumwan, Bangkok, Thailand). The MNV strains-S99 (European Virus Archive- Global, Marseille, France) was inoculated in RAW264.7 cells at a multiplicity of infection (MOI) of 1, incubated at 37 °C for 2 h in high-glucose DMEM, and replaced with low-glucose DMEM supplemented with glutamine before frozen and thawed twice, followed by centrifugation at 3500× *g* for 30 min at 4 °C according to a previous protocol [24]. Then, the supernatant was measured for viral titer by TCID50 (the median tissue culture infectious dose; the dilution of a virus required to infect 50% of a given cell culture) using the Spearman and Kärber algorithm, aliquoted and stored at −80 °C before use [24]. 

After that, 8-week-old mice were orally inoculated with 5 × 10^5^ TCID50 in 0.3 mL PBS or control (supernatant from uninfected RAW264.7 cells), and MNV in the feces was measured by reverse transcription quantitative real-time polymerase chain reaction (RT-qPCR) following a previous publication [24]. One month after gavage of MNV or control, mice (MNV negative and positive) were divided into three groups, including (i) LPS tolerance, LPS/LPS (intraperitoneal injection of 10 mg/kg LPS, *Escherichia coli* 026:B6, Sigma-Aldrich), with a repeated dose 48 h later, (ii) a single LPS stimulation (10 mg/kg intraperitoneal PBS injection followed by LPS 48 h later), and (iii) control, N/N (intraperitoneal PBS injection twice with 48 h duration between the doses). One hour after these protocols, mice were sacrificed with cardiac puncture under isoflurane anesthesia, and samples were collected (organ and blood). The stool consistency index was graded into four scores as follows; 0: normal, 1: soft, 2: loose, and 3: diarrhea, as previously published [62,63]. The cecum was washed and sonicated with the setting of pulse-on for 20 s and pulse-off for 5 s for 30 min on ice using a Sonics Vibra Cell machine (VCX 750, Sonics & Materials Inc., Newtown, CT, USA) until a homogeneous solution was obtained for the qRT-PCR procedure following a previous publication [64]. For leaky gut evaluation, 0.5 mL of 25 mg/mL fluorescein isothiocyanate (FITC)-dextran (Sigma) in sterile water was orally administered 3 h prior to sacrifice, and FITC-dextran in blood samples were measured with a fluorospectrometer (Varioskan, Thermo Fisher Scientific, San Jose, CA, USA) relative to FITC-dextran standards, as previously described [65,66,67].

### 4.5. RNA Extraction and Real-Time PCR for Gene Expression

The RNA was extracted from cells or tissues using a TRIzol Reagent (Invitrogen, Carlsbad, CA, USA) together with an RNeasy Mini Kit (Qiagen, Hilden, Germany) according to the manufacturer’s instructions. As such, 1 mg of total RNA was used for cDNA synthesis with an iScript reverse transcription supermix (Bio-Rad, Hercules, CA, USA). Quantitative real-time PCR was performed on a QuantStudio 5 real-time PCR system (Thermo Fisher Scientific, San Jose, CA, USA) using a SsoAdvanced Universal SYBR Green Supermix (Bio-Rad, Hercules, CA, United States). Expression values were normalized to Glyceraldehyde 3-phosphate dehydrogenase (*Gapdh*) as an endogenous housekeeping gene, and the fold change was calculated using the ∆∆Ct method. All primers used in this study are detailed in Table 2.

### 4.6. Statistical Analysis

The results were shown in mean ± S.E.M. All data were analyzed with GraphPad Prism6. Student’s t-test or one-way analysis of variance (ANOVA) with Tukey’s comparison test was used for the analysis of experiments with two and more than two groups, respectively. For all data sets, a *p*-value less than 0.05 was considered significant.

## 5. Conclusions

The demonstrated proteomic analysis of LPS tolerance macrophages reduced cell energy status in both mitochondrial and glycolysis activities along with anti-viral defects were supported in vitro by the extracellular flux analysis and down-regulation of some genes (*Mx1* and *Isg15*) in macrophages. Impacts of LPS tolerance in viral infection were also demonstrated in asymptomatic MNV-infected mice through diarrhea and MNV reactivation. Clinical identification of LPS tolerance might be an interesting biomarker for immune exhaustion in sepsis. 

## Figures and Tables

**Figure 1 ijms-24-01829-f001:**
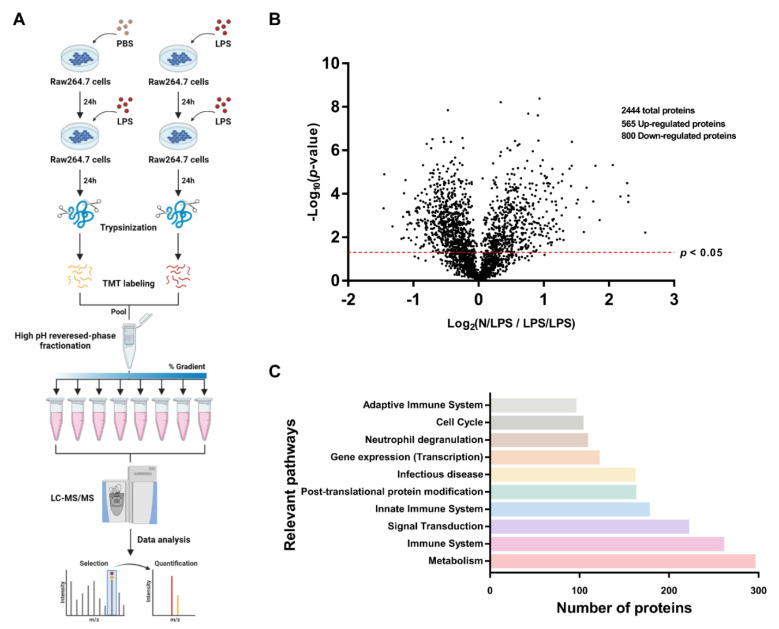
The schema of the in vitro experiments (**A**) on macrophages after the activation by (i) a single lipopolysaccharide (LPS) that started with phosphate buffer solution (PBS) followed by either LPS 24 h later (N/LPS protocol) (**A**, left side) or (ii) LPS tolerance (LPS/LPS) (**A**, right side) before performing liquid chromatography–tandem mass spectrometry (LC–MS/MS). The results are demonstrated, including the volcano plot (**B**) and the relevant pathway from the proteins as analyzed by the DAVID Bioinformatics Resources 6.8 (**C**).

**Figure 2 ijms-24-01829-f002:**
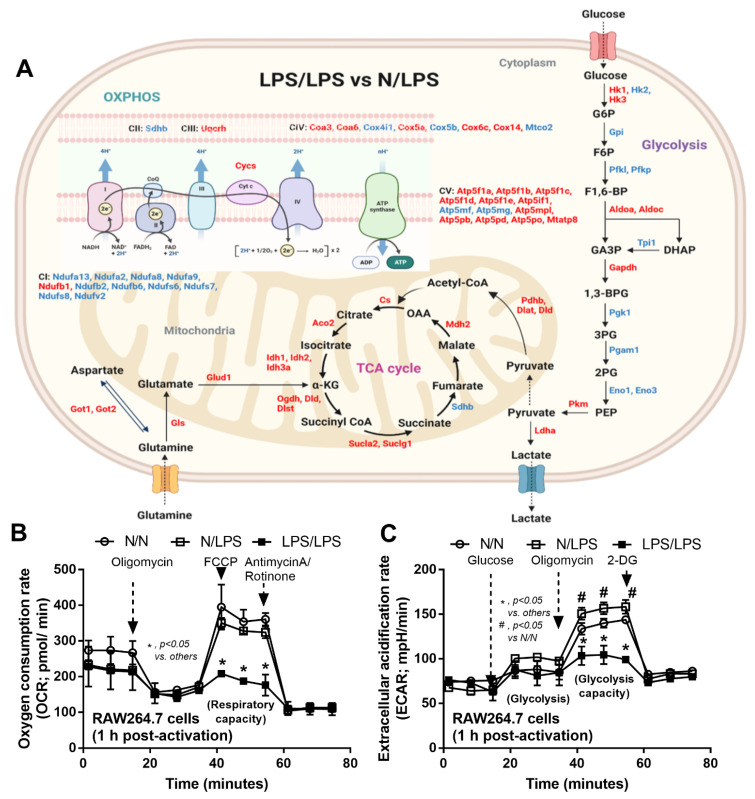
The pathway mapping of the proteomic analysis is demonstrated, focusing on the cell energy status from the protein of macrophages (RAW264.7) after lipopolysaccharide (LPS) tolerance (twice-stimulated LPS, LPS/LPS) relative to the proteins from macrophages (RAW264.7) after a single LPS stimulation (started with media and followed by LPS, N/LPS) (**A**) and the extracellular flux analysis of macrophages after N/LPS, LPS/LPS, and control N/N (twice-stimulated media), as evaluated from oxygen consumption rate (mitochondrial functions) (**B**) and extracellular acidification rate (glycolysis activity) (**C**). The protein in red- and blue-colored texts in A represent the up- and down-regulated proteins, respectively, and the independent triplicated experiments were performed for B and C. Pictures were created by BioRender.com (accessed on 10 December 2021) and the abbreviations are listed in Table 1.

**Figure 3 ijms-24-01829-f003:**
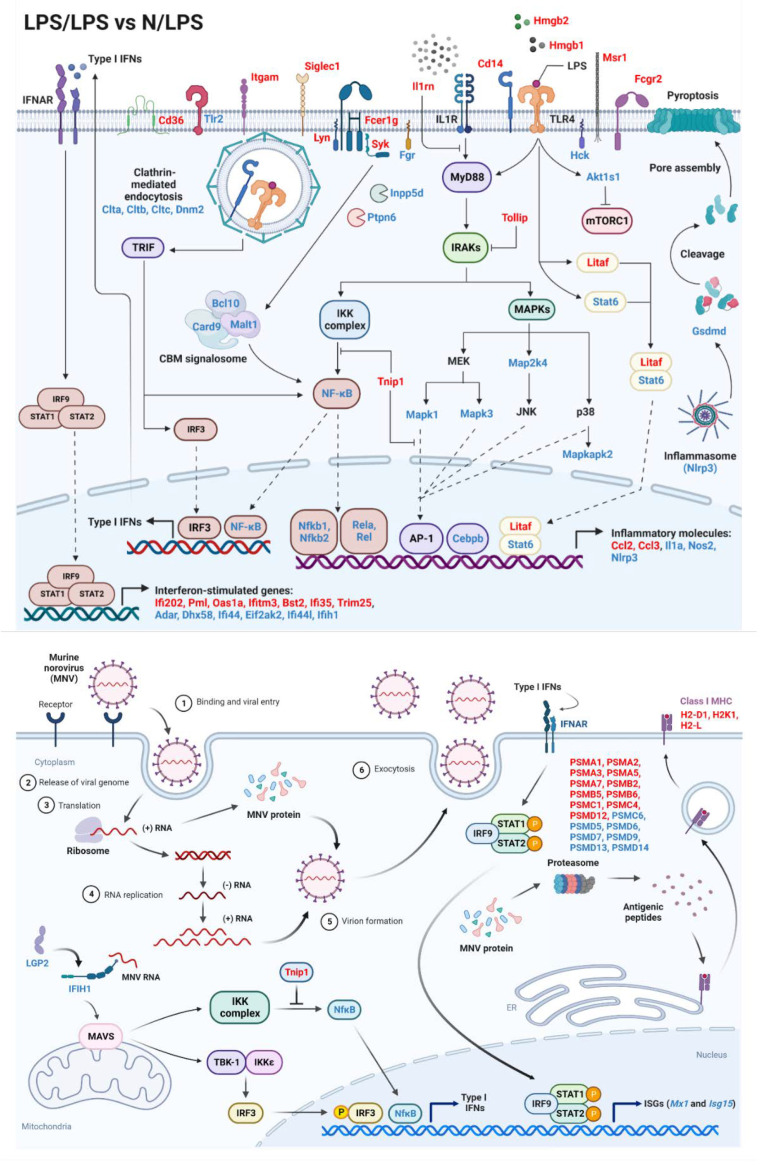
The pathway mapping of the proteomic analysis, focusing on the inflammatory signals (upper part) and viral replication pathway with antigen-presenting processes (lower part) from the protein of macrophages (RAW264.7) after lipopolysaccharide (LPS) tolerance (twice-stimulated LPS, LPS/LPS) relative to the proteins from macrophages (RAW264.7) after a single LPS stimulation (N/LPS) is demonstrated. The protein in red- and blue-colored text represents the up- and down-regulated proteins, respectively. Pictures were created by BioRender.com and the abbreviations are listed in Table 1.

**Figure 4 ijms-24-01829-f004:**
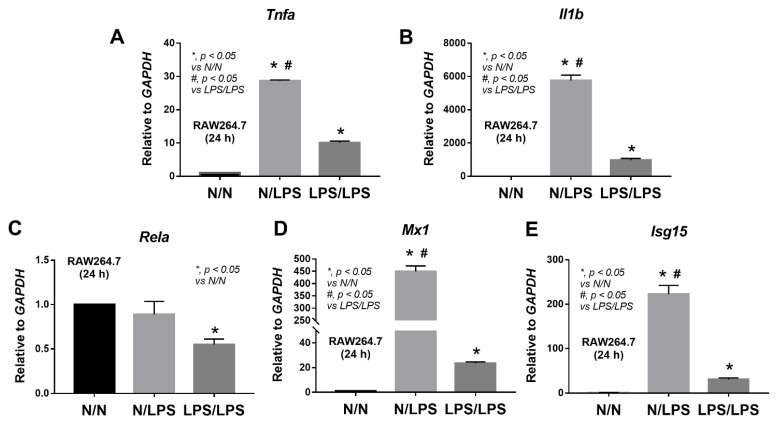
Characteristics of macrophages (RAW264.7) 24 h after the stimulation by lipopolysaccharide (LPS) tolerance (twice-stimulated LPS, LPS/LPS), a single LPS stimulation (started with phosphate buffer solution (PBS) followed by LPS, N/LPS), or control (twice-incubated PBS, N/N) as indicated by the expression of several genes, including *Tnfa* (Tumor necrosis factor alpha), *Il1b* (interleukin-1 beta), and *Rela* (the REL-associated protein for NFκB synthesis), *Mx1* (Myxovirus resistance protein 1 for viricidal effects), and *Isg15* (interferon-stimulated gene 15 for interferon synthesis) are demonstrated (**A**–**E**). Independent experiments were performed.

**Figure 5 ijms-24-01829-f005:**
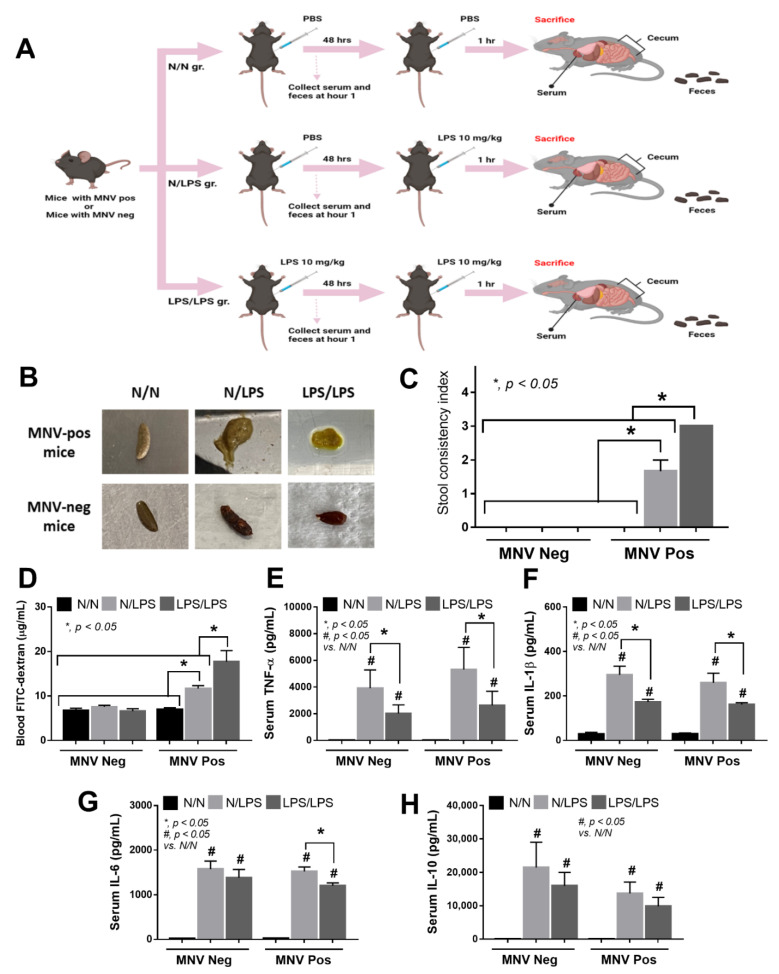
Schematic workflow (**A**) demonstrates the experimental groups, including lipopolysaccharide (LPS) tolerance, which started with an LPS injection followed by LPS (LPS/LPS); a single LPS stimulation, which started with phosphate buffer solution (PBS) followed by LPS (N/LPS); control, which started with twice-administered PBS (N/N). Experiments were performed in mice that were either positive or negative for murine norovirus (MNV). The characteristics of MNV mice, as indicated by the stool consistency index with representative fecal pictures (**A**,**B**), a gut barrier defect (FITC-dextran assay) (**C**), and serum cytokines (TNF-α, IL-1β, IL-6, and IL-10) (**E**–**H**), are demonstrated (*n* = 5–10/group).

**Figure 6 ijms-24-01829-f006:**
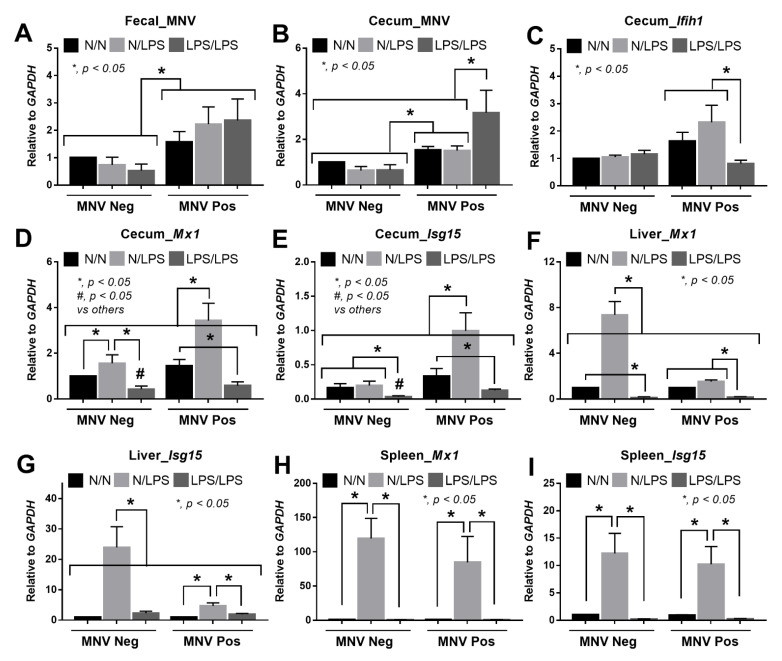
The characteristics of mice who are positive or negative for murine noroviruses (MNV) after stimulation by lipopolysaccharide (LPS) tolerance (started with LPS and followed by LPS, LPS/LPS), a single LPS stimulation (started with phosphate buffer solution (PBS) and followed by LPS, N/LPS), and control (twice-administered PBS, N/N) are demonstrated, as indicated by the fecal abundance of MNV (in feces and cecal tissue) (**A**,**B**) and the cecal responses through the expression of *Isg15* (interferon-stimulated gene 15 for interferon synthesis), *Mx1* (interferon-induced GTP-binding protein using for anti-viral activities), and *Ifih1* (interferon induced with helicase C domain 1) which encoded cytosolic RNA-sensor Melanoma differentiation-associated 5 (MDA5) (**C**–**E**) together with the possible systemic anti-viral effect through the expression of *Mx1* and *Isg15* in the liver and spleen (**F**–**I**) (*n* = 5–10/group).

**Figure 7 ijms-24-01829-f007:**
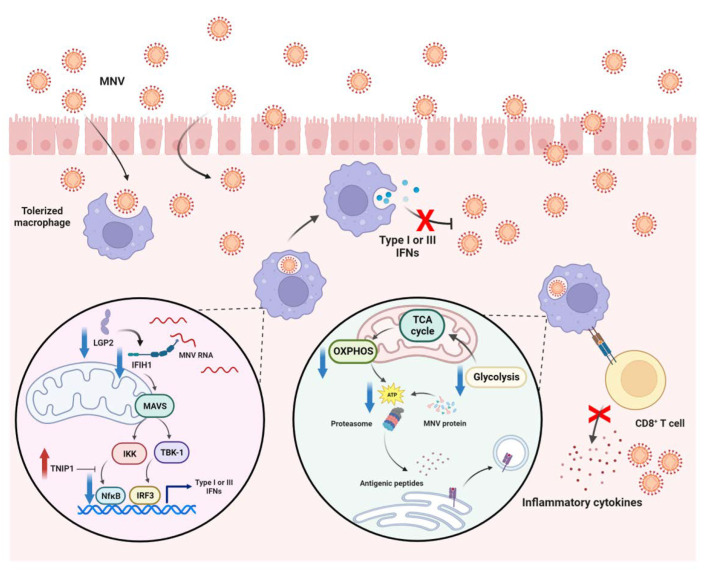
The proposed working hypothesis of the enhanced murine norovirus (MNV) infection in macrophages with lipopolysaccharide (LPS) tolerance. As such, MDA5 (Melanoma differentiation-associated 5) and LGP2 (Laboratory of genetics and physiology 2) are cytosolic RNA sensors that recognize RNA viruses (such as MNV) resulting in type I and III interferons (IFNs) using some transcription factors, such as nuclear factor kappa B (NFκB) and interferon regulatory factor 3 (IRF3). Notably, the MDA5 protein is encoded by *Ifih1* (interferon induced with helicase C domain 1) gene, and LGP2 is encoded by *Dhx58* (ATP-dependent RNA helicase DExH-box helicase 58) gene (left circle). In LPS tolerance, the reduction of several proteins, including MDA5, LGP2, NFκB (p50, p52, p65, and Rel), and proteins in the cell energy production (right cycle) with an increase in TNIP1 (TNFAIP3 Interacting Protein 1, a negative NFκB regulator) resulting in lower IFNs that enhance MNV replication. Of note, some steps of the antigen-presenting activation of CD8+ T cells require cell energy (ATP) for proteasome to generate antigenic peptides; thus, the loss of cell energy might affect CD8+ T cells’ viricidal activities.

**Table 1 ijms-24-01829-t001:** List of abbreviations.

Abbr	Name	Abbr	Name
Aco2	Aconitate hydratase, mitochondrial	Irf9	Interferon regulatory factor 9
Adar	Double-stranded RNA-specific adenosine deaminase	Isg15	Ubiquitin-like protein ISG15
Akt1S1	Proline-rich AKT1 substrate 1	Itgam	Integrin alpha-M
Aldoa	Fructose-bisphosphate aldolase A	Ldha	L-lactate dehydrogenase A chain
Aldoc	Fructose-bisphosphate aldolase C	Lgp2	Laboratory of genetics and physiology 2
Atp5f1a	ATP synthase subunit alpha, mitochondrial	Litaf	Lipopolysaccharide-induced tumor necrosis factor-alpha factor homolog
Atp5f1b	ATP synthase subunit beta, mitochondrial	Lyn	Tyrosine-protein kinase Lyn
Atp5f1c	ATP synthase subunit gamma, mitochondrial	Malt1	Mucosa-associated lymphoid tissue lymphoma translocation protein 1 homolog
Atp5f1d	ATP synthase subunit delta, mitochondrial	Map2k4	Dual-specificity mitogen-activated protein kinase 4
Atp5f1e	ATP synthase subunit epsilon, mitochondrial	Mapk1	Mitogen-activated protein kinase 1
Atp5if1	ATPase inhibitor, mitochondrial	Mapk3	Mitogen-activated protein kinase 3
Atp5mf	ATP synthase subunit f, mitochondrial	Mapkapk2	MAP kinase-activated protein kinase 2
Atp5mg	ATP synthase subunit g, mitochondrial	MAVS	Mitochondrial anti-viral signaling protein
Atp5mpl	ATP synthase subunit ATP5MPL, mitochondrial	Mdh2	Malate dehydrogenase, mitochondrial
Atp5pb	ATP synthase F(0) complex subunit B1, mitochondrial	Msr1	Macrophage scavenger receptor types I and II
Atp5pd	ATP synthase subunit d, mitochondrial	Mtatp8	ATP synthase protein 8
Atp5po	ATP synthase subunit O, mitochondrial	Mtco2	Cytochrome c oxidase subunit 2
Bcl10	B-cell lymphoma/leukemia 10	Mx1	Myxovirus resistance protein 1
Bst2	Bone marrow stromal antigen 2	Ndufa13	NADH dehydrogenase [ubiquinone] 1 alpha subcomplex subunit 13
Card9	Caspase recruitment domain-containing protein 9	Ndufa2	NADH dehydrogenase [ubiquinone] 1 alpha subcomplex subunit 2
Ccl2	C-C motif chemokine 2	Ndufa8	NADH dehydrogenase [ubiquinone] 1 alpha subcomplex subunit 8
Ccl3	C-C motif chemokine 3	Ndufa9	NADH dehydrogenase [ubiquinone] 1 alpha subcomplex subunit 9
Cd14	Monocyte differentiation antigen CD14	Ndufb1	NADH dehydrogenase [ubiquinone] 1 beta subcomplex subunit 1
Cd36	Platelet glycoprotein 4	Ndufb2	NADH dehydrogenase [ubiquinone] 1 beta subcomplex subunit 2
Cebpb	CCAAT/enhancer-binding protein β	Ndufb6	NADH dehydrogenase [ubiquinone] 1 beta subcomplex subunit 6
Clta	Clathrin light chain A	Ndufs6	NADH dehydrogenase [ubiquinone] iron-sulfur protein 6
Cltb	Clathrin light chain B	Ndufs7	NADH dehydrogenase [ubiquinone] iron-sulfur protein 7
Cltc	Clathrin heavy chain 1	Ndufs8	NADH dehydrogenase [ubiquinone] iron-sulfur protein 8
Coa3	Cytochrome c oxidase assembly factor 3 homolog	Ndufv2	NADH dehydrogenase [ubiquinone] flavoprotein 2
Coa6	Cytochrome c oxidase assembly factor 6 homolog	Nfkb1	Nuclear factor NF-kappa-B p105 subunit
Cox14	Cytochrome c oxidase assembly protein COX14	Nfkb2	Nuclear factor NF-kappa-B p100 subunit
Cox4I1	Cytochrome c oxidase subunit 4 isoform 1, mitochondrial	Nlrp3	NACHT, LRR, and PYD domains-containing protein 3
Cox5a	Cytochrome c oxidase subunit 5A, mitochondrial	Nos2	Nitric oxide synthase, inducible
Cox5b	Cytochrome c oxidase subunit 5B, mitochondrial	Oas1a	2’-5’-oligoadenylate synthase 1A
Cox6c	Cytochrome c oxidase subunit 6C	Ogdh	2-oxoglutarate dehydrogenase, mitochondrial
Cs	Citrate synthase, mitochondrial	Pdhb	Pyruvate dehydrogenase E1 component subunit beta
Cycs	Cytochrome c, somatic	Pfkl	ATP-dependent 6-phosphofructokinase, liver type
Dhx58	Probable ATP-dependent RNA helicase DHX58	Pfkp	ATP-dependent 6-phosphofructokinase, platelet type
Dlat	Dihydrolipoyllysine-residue acetyltransferase component of pyruvate dehydrogenase complex, mitochondrial	Pgam1	Phosphoglycerate mutase 1
Dld	Dihydrolipoyl dehydrogenase, mitochondrial	Pgk1	Phosphoglycerate kinase 1
Dlst	Dihydrolipoyllysine-residue succinyltransferase component of 2-oxoglutarate dehydrogenase complex, mitochondrial	Pkm	Pyruvate kinase PKM
Dnm2	Dynamin-2	Pml	Protein PML
Eif2ak2	Interferon-induced, double-stranded RNA-activated protein kinase	Psma1	Proteasome subunit alpha type-1
Eno1	Alpha-enolase	Psma2	Proteasome subunit alpha type-2
Eno3	Beta-enolase	Psma3	Proteasome subunit alpha type-3
Fcer1g	High-affinity immunoglobulin epsilon receptor subunit gamma	Psma5	Proteasome subunit alpha type-5
Fcgr2	Low-affinity immunoglobulin gamma Fc region receptor II	Psma7	Proteasome subunit alpha type-7
Fgr	Tyrosine-protein kinase Fgr	Psmb2	Proteasome subunit beta type-2
Gapdh	Glyceraldehyde-3-phosphate dehydrogenase	Psmb5	Proteasome subunit beta type-5
Gls	Glutaminase kidney isoform, mitochondrial	Psmb6	Proteasome subunit beta type-6
Glud1	Glutamate dehydrogenase 1, mitochondrial	Psmc1	26S proteasome regulatory subunit 4
Got1	Aspartate aminotransferase, cytoplasmic	Psmc4	26S proteasome regulatory subunit 6B
Got2	Aspartate aminotransferase, mitochondrial	Psmc6	26S proteasome regulatory subunit 10B
Gpi	Glucose-6-phosphate isomerase	Psmd12	26S proteasome non-ATPase regulatory subunit 12
Gsdmd	Gasdermin-D	Psmd13	26S proteasome non-ATPase regulatory subunit 13
H2-D1	H-2 class I histocompatibility antigen, D-D alpha chain	Psmd14	26S proteasome non-ATPase regulatory subunit 14
H2-K1	H-2 class I histocompatibility antigen, K-B alpha chain	Psmd5	26S proteasome non-ATPase regulatory subunit 5
H2-L	H-2 class I histocompatibility antigen, L-D alpha chain	Psmd6	26S proteasome non-ATPase regulatory subunit 6
Hck	Tyrosine-protein kinase HCK	Psmd7	26S proteasome non-ATPase regulatory subunit 7
Hk1	Hexokinase-1	Psmd9	26S proteasome non-ATPase regulatory subunit 9
Hk2	Hexokinase-2	Ptpn6	Tyrosine-protein phosphatase non-receptor type 6
Hk3	Hexokinase-3	Rela	Transcription factor p65
Hmgb1	High mobility group protein B1	Rel	Proto-oncogene c-Rel
Hmgb2	High mobility group protein B2	Sdhb	Succinate dehydrogenase [ubiquinone] iron-sulfur subunit
Idh1	Isocitrate dehydrogenase [NADP] cytoplasmic	Siglec1	Sialoadhesin
Idh2	Isocitrate dehydrogenase [NADP], mitochondrial	Stat1	Signal transducer and activator of transcription 1
Idh3a	Isocitrate dehydrogenase [NAD] subunit alpha	Stat2	Signal transducer and activator of transcription 2
Ifi202	Interferon-activable protein 202	Stat6	Signal transducer and transcription activator 6
Ifi35	Interferon-induced 35 kDa protein homolog	Sucla2	Succinate—CoA ligase [ADP-forming] subunit beta
Ifi44	Interferon-induced protein 44	Suclg1	Succinate—CoA ligase [ADP/GDP-forming] subunit alpha
Ifi44l	Interferon-induced protein 44-like	Syk	Tyrosine-protein kinase SYK
Ifih1	Interferon-induced helicase C domain-containing protein 1	Tbk1	Serine/threonine-protein kinase TBK1
Ifitm3	Interferon-induced transmembrane protein 3	Tlr2	Toll-like receptor 2
IKK	Inhibitory kappa B kinase	Tnip1	TNFAIP3-interacting protein 1
Il1a	Interleukin-1 alpha	Tollip	Toll-interacting protein
Il1rn	Interleukin-1 receptor antagonist protein	Tpi1	Triosephosphate isomerase
Inpp5d	Phosphatidylinositol 3,4,5-trisphosphate 5-phosphatase 1	Trim25	E3 ubiquitin/ISG15 ligase TRIM25
Irf3	Interferon regulatory factor 3	Uqcrh	Cytochrome b-c1 complex subunit 6, mitochondrial

**Table 2 ijms-24-01829-t002:** List of the primers.

Target Gene	Primer Sequence
Interferon-induced GTP-binding protein Mx1 (*Mx1*)	F: 5′-GATCCGACTTCACTTCCAGATGG-3′
	R: 5′-CATCTCAGTGGTAGTCCAACCC-3′
Interferon-stimulated gene 15 (*Isg15*)	F: 5′-GATCCGACTTCACTTCCAGATGG-3′
	R: 5′-GAGCTAGAGCCTGCAGCAAT-3′
NFκB/p65 (*Rela*)	F: 5′-CGTCAACTTCAAGGAAATGATGT-3′
	R: 5′-TCACAGGGTAGGAAGGCA-3′
Tumor necrosis factor alpha (*Tnfa*)	F: 5′-CTTTCTTGTTATCTTTTAAGTTGTTCTT-3′
	R: 5′-GCAGAGGTCCAAGTTCATCTTC-3′
Interleukin 1 beta (*Il1b*)	F: 5′-GGCATCAACTGACAGGTCTT-3′
	R: 5′-GCAGGATGGAGAATTACAGGAA-3′
Interferon induced with helicase C domain 1 (*Ifih1*)	F: 5′-CTTCCTCAGCCATGGTACCTCT-3′
	R: 5′-CAAGTCTTCATCAGCATCAAACTG-3′
Murine norovirus (*MNV*)	F: 5′-CACGCCACCGATCTGTTCTG-3′
	R: 5′-GCGCTGCGCCATCACTC-3′
Glyceraldehyde 3-phosphate dehydrogenase (*Gapdh*)	F: 5′-TGCACCACCAACTGCTTAGC-3′
	R: 5′-GGATGCAGGGATGATGTTCT-3′

## Data Availability

Not applicable.

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
