# Peer review of "Lipopolysaccharide Tolerance Enhances Murine Norovirus Reactivation: An Impact of Macrophages Mainly Evaluated by Proteomic Analysis"

_ijms, 2023, doi:10.3390/ijms24031829_

Round 1
Reviewer 1 Report
A very interesting and carefully conducted study containing both in vitro and in vivo elements.
In my opinion, the title of the manuscript is not appropriate, the experiments do not include a challenge infection, but a possible reactivation after LPS priming.
The results section is in part rather discussion. Here are some examples:
Lines 108-109
……supported the possible metabolic reprogramming during macrophage responses against infection [42-44].
Lines 115-118
The defect in CI of the electron transport chain might be responsible for the reduced mitochondrial function in LPS/LPS macrophages as evaluated by the oxygen consumption rate (OCR) from the extracellular flux analysis (Fig 2B)
121-122
These findings might be responsible for the reduced glycolysis in the extracellular flux analysis (Fig 2C).
Lines 126-130
This process might be an attempt to enhance the production of nicotinamide adenine dinucleotide with hydrogen (NADH) and flavin adenine dinucleotide with hydrogen (FADH2) …..
Lines 259ff
Hence, our in vivo data supported a possible viral reactivation of cecal MNV with LPS tolerance which might partly be responsible for the more severe diarrhea in LPS/LPS-activated MNV-positive mice over the MNV-negative mice (Fig 5B, C). We hypothesized that LPS tolerance might partly be responsible for the viral reactivation during sepsis together with other factors of sepsis-induced immune exhaustion.
In contrast, the discussion remains very general, e.g.
Line 292 f……. an alteration in mitochondrial function in the single LPS-stimulated cells (N/LPS), supporting previous publications [9,61-63].
Lines 374 ff
The latent MNV infection is an interesting virus to demonstrate the viral reactivation in mice because most of the viruses with latent infection in humans, especially the herpes family, do not infect in the immunocompetent rodent. The use of MNV for the viral reactivation topic is uncomplicated and further experiments focusing on the impacts of MNV in several mouse models are interesting.
M&M
Line 415. Please add LPS after 100 ng/mL
Author Response
Reviewer 1
A very interesting and carefully conducted study containing both in vitro and in vivo elements.
In my opinion, the title of the manuscript is not appropriate, the experiments do not include a challenge infection, but a possible reactivation after LPS priming.
ANS: We thank the reviewer for the comment and change the title to “Lipopolysaccharide tolerance enhances murine norovirus reactivation, an impact of macrophages as mainly evaluated by proteomic analysis”.
The results section is in part rather discussion. Here are some examples:
ANS: We thank the reviewer for the comment and move most of the discussion in the result section into the discussion section.
Lines 108-109 “…supported the possible metabolic reprogramming during macrophage responses against infection [42-44].”
ANS: We thank the reviewer for the comment, cut this sentence in result section and move to the new discussion as following “The most abundant proteins in LPS tolerance (LPS/LPS) macrophages compared with the single LPS-activated macrophages (N/LPS) from the DAVID Bioinformatics tools were the members of metabolic processes indicating the possible metabolic reprogramming during macrophage responses against infection supported previous publications [42-44].”.
Lines 115-118 “The defect in CI of the electron transport chain might be responsible for the reduced mitochondrial function in LPS/LPS macrophages as evaluated by the oxygen consumption rate (OCR) from the extracellular flux analysis (Fig 2B)” line 121-122 “These findings might be responsible for the reduced glycolysis in the extracellular flux analysis (Fig 2C).” Lines 126-130 “This process might be an attempt to enhance the production of nicotinamide adenine dinucleotide with hydrogen (NADH) and flavin adenine dinucleotide with hydrogen (FADH2)” …..
ANS: We thank the reviewer for the comment. We cut these parts and replace with these following sentences in the results “In parallel, there was a reduction in mitochondrial function and glycolysis activity of LPS/LPS macrophages compared with N/LPS cells as indicated by the oxygen consumption rate (OCR) from the extracellular flux analysis (Fig 2B) and the decreased hexokinase 2 (Hk2) and phosphofructokinase (liver- and platelet-type) (Pfkl and Pfkp) (the important rate-limiting glycolytic enzymes) (Fig 2A, right side), respectively. Additionally, there was higher expression of several enzymes in glutamine metabolism (Got1, Got2, Gls, and Glud1) in LPS/LPS macrophages over the N/LPS cells (Fig 2A, left lower part).”.
Lines 259ff “Hence, our in vivo data supported a possible viral reactivation of cecal MNV with LPS tolerance which might partly be responsible for the more severe diarrhea in LPS/LPS-activated MNV-positive mice over the MNV-negative mice (Fig 5B, C). We hypothesized that LPS tolerance might partly be responsible for the viral reactivation during sepsis together with other factors of sepsis-induced immune exhaustion.”
ANS: We thank the reviewer for the comment and move these parts to the discussion.
In contrast, the discussion remains very general, e.g.
Line 292 f……. an alteration in mitochondrial function in the single LPS-stimulated cells (N/LPS), supporting previous publications [9,61-63].
ANS: We thank the reviewer for the comment and put the modified sentences from the results of the previous version of the manuscript in the new discussion (Line 270-296 in the new version of manuscript).
Lines 374 ff “The latent MNV infection is an interesting virus to demonstrate the viral reactivation in mice because most of the viruses with latent infection in humans, especially the herpes family, do not infect in the immunocompetent rodent. The use of MNV for the viral reactivation topic is uncomplicated and further experiments focusing on the impacts of MNV in several mouse models are interesting.”
ANS: We thank the reviewer for the comment and move the sentences from the result to the new discussion as following “Here, our in vivo data supported a possible viral reactivation of cecal MNV with LPS tolerance which might partly be responsible for the more severe diarrhea in LPS/LPS-activated MNV-positive mice over the MNV-negative mice. We hypothesized that LPS tolerance might partly be responsible for the viral reactivation during sepsis together with other factors of sepsis-induced immune exhaustion.”
M&M Line 415. Please add LPS after 100 ng/mL
ANS: We thank the reviewer for the comment and add it accordingly.
Reviewer 2 Report
This manusript describes the proteomic analysis in lipopolysaccharide (LPS) tolerance macrophages compared with single LPS-stimulated cells using RAW264.7 as well as the LPS tolerance effect in murine norovirus-infected versus control mice. The Authors used a wide range of biological tests in order to perform the experiments including: extracellular flux analysis, animal model, RNA extraction and real-time PCR for gene expression. In my opinion, the experiments were well planned and the results are clearly presented. I would only add more recent references. Only 39 from 91 citations are from 2020 and beyond. In my opinion, at least half of the citations in the work should be from the last three years. After completing this remark, the work can be published in Int. J. Mol. Sci.
Author Response
Reviewer 2
This manuscript describes the proteomic analysis in lipopolysaccharide (LPS) tolerance macrophages compared with single LPS-stimulated cells using RAW264.7 as well as the LPS tolerance effect in murine norovirus-infected versus control mice. The Authors used a wide range of biological tests in order to perform the experiments including: extracellular flux analysis, animal model, RNA extraction and real-time PCR for gene expression. In my opinion, the experiments were well planned and the results are clearly presented. I would only add more recent references. Only 39 from 91 citations are from 2020 and beyond. In my opinion, at least half of the citations in the work should be from the last three years. After completing this remark, the work can be published in Int. J. Mol. Sci.
ANS: We thank the reviewer for the comment and cut most of the non-update references (cut out more than 20 references). Only a few of references that published before 2019 are still needed to be mentioned as the necessary references for the specific sentence.